# Encoding Stability into Laser Powder Bed Fusion Monitoring Using Temporal Features and Pore Density Modelling

**DOI:** 10.3390/s22103740

**Published:** 2022-05-14

**Authors:** Brian G. Booth, Rob Heylen, Mohsen Nourazar, Dries Verhees, Wilfried Philips, Abdellatif Bey-Temsamani

**Affiliations:** 1imec TELIN-IPI, Ghent University, 3000 Leuven, Belgium; mohsen.nourazar@ugent.be (M.N.); wilfried.philips@ugent.be (W.P.); 2Flanders Make, 3920 Lommel, Belgium; rob.heylen@flandersmake.be (R.H.); dries.verhees@flandersmake.be (D.V.); abdellatif.bey-temsamani@flandersmake.be (A.B.-T.)

**Keywords:** melt pool monitoring, laser powder bed fusion, keyhole pores, lack-of-fusion pores, temporal features

## Abstract

In laser powder bed fusion (LPBF), melt pool instability can lead to the development of pores in printed parts, reducing the part’s structural strength. While camera-based monitoring systems have been introduced to improve melt pool stability, these systems only measure melt pool stability in limited, indirect ways. We propose that melt pool stability can be improved by explicitly encoding stability into LPBF monitoring systems through the use of temporal features and pore density modelling. We introduce the temporal features, in the form of temporal variances of common LPBF monitoring features (e.g., melt pool area, intensity), to explicitly quantify printing stability. Furthermore, we introduce a neural network model trained to link these video features directly to pore densities estimated from the CT scans of previously printed parts. This model aims to reduce the number of online printer interventions to only those that are required to avoid porosity. These contributions are then implemented in a full LPBF monitoring system and tested on prints using 316L stainless steel. Results showed that our explicit stability quantification improved the correlation between our predicted pore densities and true pore densities by up to 42%.

## 1. Introduction

Customised production is a growing trend in additive manufacturing, especially in the generation of smaller series of products with complex geometries. As a result of this trend, laser powder bed fusion (LPBF) is increasingly used due to its suitability in creating complex, custom metal parts [1]. LPBF creates metal parts by melting metal powder with a laser in our pattern of choice. These patterns are derived from a computer-aided design (CAD), which easily allows for the modelling of flexible and complex designs. As a result, LPBF has seen significant use in the production of medical implants, machine components (e.g., turbine blades), and aeronautical parts [2].

Quality assurance is the main challenge in LPBF. During an LPBF build process, various defects can occur, which impact the strength and structural integrity of the produced part [3,4]. These defects can be the result of printing instabilities brought on by, for example, variations in the material properties of the metal powder, heat accumulating in the printed part, and disturbances in the build environment (e.g., changes in humidity and air flow) [5,6]. One such defect type is *porosity*: the introduction of pores (or voids) in the printed parts due to unstable printing conditions. Pores reduce the structural strength of a printed part and make the parts more susceptible to cracking and fatigue. To reduce porosity, the current approach is to test the printed specimens using destructive or non-destructive testing, then adjusting the print settings by trial and error based on the test findings. These repeated build attempts lead to high scrap rates, which inflate the costs involved in LPBF, and these costs act as a barrier to further uptake of the technology [7].

In order to reduce the scrap rates in LPBF, we require the ability to reduce porosity by creating and maintaining a stable printing process, even in a constantly changing environment. In order to adjust build parameters on-the-fly, the state of the LPBF process needs to be actively monitored, and any instabilities need to be communicated to the 3D printer in real-time [4,6].

Various approaches have been proposed to monitor the stability of the LPBF process, with differences in the choice of sensor to sense the printing process [8,9,10], the choice of analysis to be performed [11,12,13], and the signals communicated to the LPBF printer [14,15,16]. The use of optical sensors has been preferred in these monitoring systems with the trend being towards more detailed sensing: initially, photodiodes were the sensor of choice [12,17,18], but more recently, cameras in the visual [19,20] or infrared spectrum [21,22] have seen increasingly use. The use of cameras provides valuable spatial information on the melt pool and spatters, but requires more elaborate processing compared to photodiodes.

The processing of the melt pool video typically consists of two stages: feature extraction and the translation of features into useful printer control signals. For feature extraction, classical hand-designed features such as melt pool geometry [9] or image texture features [23] have been used due to the ease of computation and interpretation. More recently, deep learning approaches have also been applied to learn features from individual video frames [16,24,25]. While more descriptive, these deep learning features come with a higher computational cost. Regardless of the features chosen, they serve as inputs to the computation of either anomaly scores [9,11,14] or to estimate printer parameters (e.g., laser speed and power) [10,13,26]. While anomaly scores capture how much the current LPBF behaviour deviates from nominal, they do not describe in what way the current LPBF behaviour differs. As a result, anomaly scores are typically used offline to highlight results that require further investigation. Meanwhile, the estimation of printing parameters can highlight instability in the printing process if they differ from the parameters that are actually being used. This “estimation error” also provides an intuitive control signal for online intervention in the LPBF process (e.g., parameters are adjusted in an equal, but opposite fashion to the reported estimation errors).

While these existing works show promise in stabilising LPBF behaviour, features explicitly expressing LPBF stability do not appear in these LPBF monitoring systems. Specifically, feature extraction is entirely performed in the spatial domain. The temporal domain is more naturally suited to capture process stability, but it is generally ignored with the exception of some simple temporal denoising (e.g., averaging multiple video frames prior to feature extraction) [8,15]. Such denoising tends to remove temporal fluctuations, which may indicate printing instability. Additionally, the way in which print quality is summarised can also influence print stability through the form of online control it produces. Existing techniques, such as the estimation printing parameters or anomaly scores, would tell us about unstable printing behaviour, but would not directly tell us whether these instabilities have led to printing defects. Thus, these control signals can be over-conservative, leading to regular printer interventions, even when there is a low likelihood of defects occurring. The frequency of these changes in printing parameters may introduce unforeseen fluctuations in the printer or the build process.

We hypothesise that the stability of the LPBF process can be improved by explicitly incorporating the concept of stability into the monitoring system itself. To test this hypothesis, we propose two novel contributions:*Introducing temporal features* into LPBF monitoring to summarise the behaviour of the LPBF process over time. These features allow for the direct measurement of print stability.*Defect density regression*: We introduce a neural network model that uniquely links video features to the pore density estimates collected from post-print X-ray computed tomography (CT) imaging. With this model, we aim to reduce the number of printer control actions down to only those that are necessary to avoid the creation of defects, thereby increasing stability in the printer control.

We integrate these contributions into an LPBF monitoring platform (Section 2) and test it on parts printed with 316L stainless steel (Section 3). We show that our contributions increase the correlation between our printer control signals and the density of observed defects. The implications of these results are then discussed (Section 4) prior to concluding (Section 5).

## 2. Materials and Methods

Figure 1 presents an overview of our proposed LPBF monitoring system. A high-speed video camera is used to collect data on the current status of the LPBF process. This camera is installed off-axis: it is fixed with respect to the build plate and does not follow the motion of the laser. This has the advantage of not requiring any machine modifications or changes in the optical path of the laser, thus allowing the monitoring system to be easily installed as an add-on. From the video, spatial features are calculated to capture information related to the LPBF printing process. For the first time, we applied temporal analysis of these spatial features to assess melt pool stability. Both the temporal and spatial features are then used as the input to a machine learning regression module to predict the density of the created pores. Specifically, we focus here on both lack-of-fusion and keyhole pore densities as the estimates of interest. CT images of printed parts were used to estimate these pore densities and to train the regression model. We envision that these pore density estimations could be used as online control signals that can be sent to the LPBF printer. Each item in the system is described further below.

### 2.1. Data Collection

#### 2.1.1. LPBF Machine, Material, and Settings

A 3D Systems ProX DMP320 LPBF machine was used to 3D print the parts examined in this study. The machine was equipped with the Materialise Control Platform (MCP) to enable a variety of experimental designs. The laser source of the LPBF machine was a ytterbium fibre with a wavelength of 1064 nm and had a spot size of 75 µm. Each LPBF print was carried out under an argon atmosphere using 316L stainless steel. The powder particle size of the stainless steel ranged between 20 and 50 µm. Each layer of the printed part was 30 µm thick, and a hatching distance of 100 µm was used. Once printed, all parts were cut from their base plates using wire electrical discharge machining (wire-EDM).

As part of the data collection, we varied the LPBF printing parameters layer-by-layer to encourage either (a) nominal print behaviour, (b) the creation of keyhole pores (where gasses are trapped in the molten metal, leading to voids in the printed part), or (c) the creation of lack-of-fusion pores (where the metal powder does not fully melt and fuses with the rest of the printed part). By introducing these variations on purpose, we can distinguish different pore types and, thus, obtain reasonable ground truth pore labels with which to train our machine learning models.

#### 2.1.2. Camera Monitoring System

A Mikrotron EoSens 3CL camera was used to record the LPBF printing process. The camera was installed as shown in Figure 2: above the build plate at an angle of $25^{\circ }$ from the vertical build axis. The camera operated at 20,000 frames per second, producing 8 bit monochrome images at a resolution of 120×120 pixels. This camera was combined with a 25 mm-focal-length lens which, in our printer, resulting in a field of view of 25.4×25.4 mm, with each pixel covering an area of 212×212 microns in size.

In order to protect the camera from being blinded by the laser source directly, a short-pass filter with a cut-off threshold of 975 nm was added to the camera. Taking the sensitivity of the camera into account, the monitoring system captured radiations between 350 nm and 975 nm.

The camera images were recorded together with the (x,y)-position of the laser on the build area. These laser positions are set points available in the MCP.

#### 2.1.3. CT Imaging System

A FleXCT micro CT imaging system was used to generate 3D X-ray CT images of the printed parts. Each part was imaged in full with a resolution of 10×10×10 µm. These images provided ground truth information on the porosity of the printed samples.

### 2.2. Optical Feature Extraction

To predict the pore densities described earlier, multiple optical features were extracted from the high-speed video. To show the generality of our contributions, a broad range of spatial features was selected based on their popularity in previous LPBF monitoring and industrial inspection works. Based on these spatial features, we added new temporal features that measured the stability in the LPBF printing process. Each of these features are described below. In the following, I(x,y,t) represents the intensity captured by the camera at pixel location (x,y) and time *t*. An example frame is shown in Figure 3a. We further assumed that a camera calibration step was performed to allow for the mapping of the printer’s coordinate system to that of the video camera [27]. As a result, we have the laser’s location in pixel coordinates as (cx(t),cy(t)), also parameterised by the time parameter *t*. Finally, we have the frame-by-frame laser displacement as v(t)=(cx(t)−cx(t−Δt),cy(t)−cy(t−Δt)). These are shown for a given time point in Figure 3b.

#### 2.2.1. Melt Pool Area

We segmented the melt pool by simple grey-value thresholding (threshold τ) and connected component analysis [28]. This is possible because the melt pool is much hotter than its surroundings, and hence no, sophisticated method is needed. In the following, the binary-valued function S(x,y,t) takes the value of 1 if pixel (x,y) is above the grey-value threshold τ. Typically, this binary segmentation consists of multiple connected components, with one of them containing (cx(t),cy(t)). We denote the segmentation of this latter component as $S_{m}\left(x,y,t\right)$, a binary-valued function that equals 1 in all pixels of the melt pool at time *t*. Let $A_{m}\left(t\right)$ denote the melt pool area at time *t*:(1)$$A_{m}\left(t\right)=\sum_{\left(x,y\right)}S_{m}\left(x,y,t\right).$$

The melt pool area should remain constant over time during a stable printing process [9,12,13,17,20,29,30,31,32]. If the melt pool gets larger than nominal, this could be evidence for a keyhole pore. If the melt pool gets smaller than nominal, this can be evidence for a lack-of-fusion pore.

#### 2.2.2. Melt Pool Width–Length Ratio

In a similar vein to the melt pool area, the shape of the melt pool is frequently used to characterise LPBF printing behaviour. This shape is often characterised by the ratio between the melt pool’s length $\ell_{\|}\left(t\right)$ along the direction of the laser’s displacement and the melt pool width $\ell_{⊥}\left(t\right)$ perpendicular to the direction of laser displacement [11,12,13,17,20,33,34,35,36]. A large melt pool length, or a low width–length ratio, is an indication of delayed melt pool solidification due to excess laser energy, which in turn may indicate the presence of keyhole pores.

To compute the length of the melt pool, we made use of the melt pool segmentation $S_{m}\left(t\right)$ described above and the laser displacement vector v(t). To calculate the melt pool length, the coordinates of each pixel in the melt pool are projected onto the laser direction vector v(t). Subsequently, the length of the melt pool is then equal to the range of these scalar projections:(2)$$\ell_{\|}\left(t\right)=\max_{\left(x,y\right)}\left(z\left(x,y,t\right)\right)-\min_{\left(x,y\right)}\left(z\left(x,y,t\right)\right),$$
where z(x,y,t) is the scalar projection of coordinate (x,y) in $S_{m}\left(t\right)$. The melt pool width $\ell_{⊥}\left(t\right)$ is computed similarly by replacing the laser displacement vector v(t) with the vector perpendicular to it.

#### 2.2.3. Amount of Spatter

Spatter refers to droplets of molten metal that are ejected from the melt pool during an LPBF print. Recently, the amount of spatter exiting the melt pool has become a feature of interest, with more spatter being a sign of higher laser energy [37].

Typically, spatters are also hotter than the surrounding material and can be found by thresholding. In this paper, we segmented both spatters and the melt pool using the same threshold τ. In the following, $S_{s}\left(x,y,t\right)$ is a binary-valued function equalling 1 if (x,y) is in a spatter region at time *t*. In our approach $S_{s}\left(x,y,t\right)=S\left(x,y,t\right)\backslash S_{m}\left(x,y,t\right)$. The amount of spatter, $A_{s}\left(t\right)$, is then computed as
(3)$$A_{s}\left(t\right)=\sum_{\left(x,y\right)}S_{s}\left(x,y,t\right).$$

#### 2.2.4. Number of Spatters

The number of spatters may, like the amount of spatter, be an indication of a defect occurring. The number of spatters is obtained by counting the number of high-intensity regions in the thresholded image *S* and subtracting one (for the melt pool).

#### 2.2.5. Melt Pool Intensity

The intensity of the melt pool is often used as a surrogate measure for the melt pool’s temperature [9,12,13,15,17,20,31,34]. We define melt pool intensity as the average intensity within the melt pool area:(4)$${\hat{I}}_{m}\left(t\right)=\frac{1}{A_{m}\left(t\right)}\sum_{\left(x,y\right)}I\left(x,y,t\right)S_{m}\left(x,y,t\right).$$

#### 2.2.6. Spatter Direction

Recently, Ji and Han [38] argued that the direction in which spatters travel may be a useful feature for identifying pore creation. To characterise spatter direction with a consistent number of features, we followed their approach and made use of a combination of the polar [39] and Radon transforms [40]. In effect, the polar transform was used to generate a small number of angular zones similar to the ones shown in blue in Figure 3c. These zones were then centred at the laser location (cx(t),cy(t)) and rotated to align with the laser displacement vector, v(t), shown in blue in Figure 3b. Once in place, the Radon transform was used to integrate image intensities over each angular zone. The sums from each zone were then used as features that roughly captured the directions of spatter. As we employed a faster camera than the one used by Ji and Han (20,000 fps vs. 50 fps), we were able to capture spatters using a single ring of angular zones as opposed to the 5 used by Ji and Han [38].

#### 2.2.7. Image Texture Features

The Histogram of Oriented Gradients (HOG) feature detector has been used in melt pool analysis before [10] to capture overall image texture. Features such as the HOG have proven to be useful in other industrial inspection problems [23] and capture more complex image cues than the other features previously mentioned. Therefore, we included them here to capture melt pool texture and the relations between spatters and the melt pool.

For the HOG detector, we computed image gradients ∇I(x,y,t) using finite differences. Traditionally, these gradients are binned based on their orientation to create a histogram. However, in our context of melt pool monitoring, we made our texture features invariant to the direction of laser travel. To do so, we measured the angle between the gradient vector, ∇I(x,y,t), and the laser displacement vector, v(t). This angle is computed as
(5)$$\theta \left(x,y,t\right)=\mathrm{acos}\left(\frac{\nabla I(x,y,t)}{∥\nabla I(x,y,t)∥}\cdot \frac{v(t)}{∥v(t)∥}\right).$$

These gradient angles are then binned to create our histogram of oriented gradients. Furthermore, unlike the traditional HOG, which creates a histogram for each 16×16 pixel block, we used a single histogram for the whole image. This was done to (a) keep the number of features at a manageable level and (b) because the melt pool video does not exhibit a large amount of image texture.

### 2.3. Temporal Features

We propose new temporal features for the direct measurement of melt pool stability. These temporal features were computed from each of the spatial features described earlier in this section. The temporal variance of each of the spatial features was used to estimate LPBF print stability over a chosen time range.

For a time-varying feature f(t), we define its temporal mean and variance as
(6)$$\mu_{t}\left(f;t\right)=\frac{1}{K}\sum_{t^{\prime }=t-K}^{t}f\left(t^{\prime }\right),$$
(7)$$\sigma_{t}^{2}\left(f;t\right)=\frac{1}{K}\sum_{t^{\prime }=t-K}^{t}{\left(f\left(t^{\prime }\right)-\mu_{t}\left(f;t\right)\right)}^{2},$$
where *K* is the number of video frames in our temporal window of interest. In this work, one temporal variance feature was computed for each of the spatial features described earlier. We expect that larger-than-nominal temporal variances would indicate instability in the printing process and a heightened potential to introduce pores.

### 2.4. Pore Density Measurement

A main contribution of this work is to directly predict the creation of pores from the high-speed video features described earlier. To achieve this goal, we employed a supervised learning approach to perform regression between the extracted video features and pore densities. To generate training data for this task, we introduced porosity in layers of the printed part using off-nominal printer settings. These layers can be identified on the CT image of the printed part, and their corresponding pores are used to define pore densities.

To extract pore densities from the CT image, the following procedure was followed (Figure 4). First, the CT image was denoised with a 3D total variation method to minimise the effect of imaging artefacts (e.g., scatter) and to improve the subsequent pore segmentation accuracy [41]. A binary segmentation mask was then obtained by comparing the CT image with a global threshold obtained using Otsu’s method. Voxels below the threshold were labelled as belonging to pores. All segmented pore voxels inside the printed object’s CT image were aggregated over each print layer. This gives a per-layer measure for the number of pore voxels. A cube root was applied to this number of voxels to obtain a quantity that scales linearly with the dimensions. We refer to this measure as the linear pore density (LPD) and used these measurements as target values in training the machine learning model.

Separate LPD estimates were generated for both lack-of-fusion and keyhole pores. The lack-of-fusion pores arise where they are created, so the LPD of the given layer was used as its lack-of-fusion pore density estimate. Meanwhile, keyhole pores are known to appear below the print layer that creates them. Therefore, the keyhole LPD for a given layer was computed as the sum of the LPD values in the 20 print layers below the given layer [42]. Because of these definitions, it is important to induce these different pore types in separate regions of the printed part when preparing your training data (i.e., layers with nominal laser settings should be printed between layers that introduce lack-of-fusion or keyhole pores).

### 2.5. Pore Density Estimation

To predict pore densities from the high-speed video, a supervised learning approach was chosen based on a neural network. We generated a training set by pairing the video features from a layer with the corresponding LPDs described in the previous subsection. To improve the optimisation of the neural network parameters, the input features were normalised so that each had a unit mean. The input dimension is equal to the number of high-speed video features, while the output dimension is 2 (LPD predictions for lack-of-fusion and keyhole pores).

Several neural network architectures were considered, including both convolutional, pooling, and fully connected layers, and we investigated several transfer functions. Preference was given to simpler networks as (a) we value future real-time performance and (b) we do not expect a very high complexity in this inference task and want to avoid overtraining or memorisation of specific data points. After several iterations, we decided on a simple structure corresponding to a two-layer perceptron with 164 neurons in the hidden layer, a sigmoid transfer function in the hidden layer, and a rectified linear unit (ReLu) transfer function in the output. Training was performed in Keras for 100 epochs with the Adam optimiser and a mean absolute error loss function.

### 2.6. Experimental Setup

#### 2.6.1. Description of Object and Printing

For the experimental validation, the test object shown in Figure 5a was printed. The design consists of a main cylinder (diameter 5 mm and height 19 mm) with several smaller cylinders protruding at angles of 45 degrees. These smaller cylinders allow easy alignment with CT reconstructions, taking into account possible shrinking. A squared non-symmetrical base plate was added as this section will get partially lost while detaching from the build plate. The markings seen on top are used to identify the object. The object was hatched in a parallel back-and-forth pattern, with a hatching distance of 100 microns and a rotation of 67 degrees between layers.

The bulk of the object was printed with nominal parameters for laser speed and power. Deviations from these nominal parameters were introduced in 14 sections. Every deviating section consisted of three consecutive layers where the 25 central lines had off-nominal laser parameters. This resulted in overlapping zones with 1, 2, and 3 deviating layers, as illustrated in Figure 5c. The settings for the different off-nominal sections are listed in Table 1 along with the LPD measurements computed from the CT scan (see Section 2.5). In between the off-nominal sections, we printed at least 1 mm of nominal layers, thereby avoiding that pores from one section are confused with another section.

The full design of the test object was developed using the Materialise Build Processor SDK, which enabled us to introduce local, off-nominal, laser settings into the print job file.

#### 2.6.2. Data Preprocessing and Model Training

Once printed, the test object was moved from the build plate and CT scanned. The resulting CT image was aligned to its CAD model using a least-squares fitting procedure. The video data were also synchronised to the CAD model, as mentioned in Section 2.1.2. The CT image and recorded video were then processed as described earlier. To avoid border effects, which introduce porosity of unknown type, pore densities were computed only on the central cylindrical core of the object (radius = 2 mm).

The data from the test object were split into a training and a testing set by vertically “slicing” the object in two. All data points from one half of the object (i.e., the data coming from laser coordinates with negative *x* values) were used for training, while the data points from the other half of the object were used for testing. Since the object is symmetric in the horizontal plane and since the hatching direction constantly alternates, the same printing behaviours are expected to be observed in both the training and test sets.

For the calculation of optical features, the segmentation threshold τ was empirically set to 40, while the length of the sliding window for the temporal variance calculations, *K*, was empirically set to 10 frames (or 0.5 ms). Additionally, the number of bins in the HOG feature detector was empirically set to 9, while the angular zones used to measure spatter direction (Figure 3c) were empirically set to $60^{\circ }$ each with a thickness of 8 pixels. This results in the set of input features listed in Table 2.

We compared our proposed monitoring framework with a baseline approach similar to that of Zhang et al. [13]. In their work, similar spatial image features were used to predict laser power settings, with incorrect predictions being potential indicators of porosity. Since both laser speed and power significantly influence porosity, we extended the work of Zhang et al. to predict both laser parameters. In the case of laser speed, we predicted the inverse laser speed instead in order to have both predicted values be linearly proportional to the energy density output by the laser. To better isolate the impact of our proposed contributions, we used our spatial features and neural network to accomplish the same task as Zhang et al., then measured the impact of adding our temporal features and pore density estimations to this framework. With the exception of the model training in Keras, all other LPBF monitoring software was implemented in Quasar [43].

## 3. Results

### 3.1. Temporal Features

Our proposed monitoring system was tested both with and without the temporal features, as was the baseline monitoring system of Zhang et al. that predicts laser speed and power. In Figure 6, the layer-averaged predictions of the neural network are plotted for these four different scenarios (with and without temporal features for the baseline system and ours). Layer averages were obtained by averaging the predictions for all video frames in a single layer that correspond to the laser-on signal. As the contours of the object are always printed at nominal settings, a slight bias might be introduced in these averages when considering error layers. For each graph, the target values for the 14 off-nominal settings in Table 1 are indicated with dots. All other layers are nominal, corresponding to target values of 1.0 (for system settings targets) or 0.0 (for pore targets).

In the first scenario with the baseline system, the settings of the laser power and inverse laser speed were predicted from only the spatial features of each individual video frame. In the left image of Figure 6a, it can be seen that the predicted energy and inverse speed correspond fairly to the ground truths, although a significant bias can be observed in the inverse speed target. At the off-nominal layers, clear peaks can be observed in the corresponding predicted values, which are proportional to the deviation from nominal. However, the off-nominal, high-laser-speed settings between Layers 400 and 500 were poorly estimated. A fast oscillating behaviour was also observed in the inverse speed estimates, with a layer frequency that correlates with the inter-layer rotation of the hatching strategy ($67^{\circ }$ per layer). This result indicates a sensitivity to hatching and laser direction.

The second scenario with the baseline system, visible in the right image of Figure 6a, repeated the experiment, but with the addition of the temporal features described in Section 2.3. The predictions showed a notable improvement: the bias and directional sensitivity were lessened, and the predicted values at the error layers were closer to the target values. This gives a qualitative confirmation that the inclusion of temporal features can significantly enhance the predictive capabilities of the network.

Figure 6b shows the pore density predictions from our proposed monitoring system, with and without the use of temporal features. Similar conclusions apply here: a bias and oscillating behaviour was present, which was lessened by the inclusion of temporal features, and the predictions showed a clear and proportional correspondence to the targets at the error layers. The inclusion of temporal features showed a large qualitative improvement in the predictions of the network.

Quantitative results were also obtained by calculating the correlation coefficients between the predictions and the ground truth values on the test set for the four different scenarios. These coefficients are shown in Table 3 and confirm several of the conclusions derived above. A clear increase in correlation can be observed when temporal features are added, in all prediction measures. The increases were most noticeable for laser speed and keyhole pore density predictions (20.7% and 20.6% increases, respectively).

Figure 7 shows an example of the improvements seen when using temporal features when printing a layer with higher-than-nominal laser speed (Section ID = 11 in Table 1). This printer setting leads to the lack-of-fusion pores seen in the red oval in Figure 7a. Without using temporal features, the laser speed estimate (left-bottom of Figure 7b) and the lack-of-fusion pore density estimate (left-top of Figure 7c) underestimated the extent of the problem. When including the temporal features, the predicted laser speed significantly increased (shown as a decrease in inverse laser speed in the right-bottom of Figure 7b) as did the predicted lack-of-fusion porosity (top-right of Figure 7c). Both estimates based on temporal features agreed better with the porosity seen on the CT image.

### 3.2. Pore Density Estimation

We hypothesised that direct predictions of pore densities would correlate more strongly with part quality than predictions of the printer’s laser settings. To test this hypothesis, we computed the coefficient of multiple correlations between predicted outputs and the measured pore densities. These correlation coefficients are presented in Table 4. We observed that predicting pore densities directly correlated significantly better to actual pore densities than using laser speed and power settings (42.8% better for lack-of-fusion pores and 24.8% better for keyhole pores).

To understand the reason why pore density prediction works better as a print quality indicator than laser settings, we present results in Figure 8 for a layer printed with higher-than-nominal laser power (Section ID = 9 in Table 1). The CT image in Figure 8a shows no porosity within the red oval in this layer, and the magnitude of high keyhole pore density predictions in Figure 8c generally aligns with the lack of porosity observed. Conversely, the predicted laser power, while well-predicted, is shown as being abnormally high throughout this region. This laser power prediction clearly does not align well with the amount of porosity observed on the CT image. Thus, a disconnect exists between the off-nominal laser setting and the amount of porosity observed.

## 4. Discussion

The hypothesis for this study was that porosity could be more accurately predicted by taking the stability of the LPBF process explicitly into account. We considered stability in two respects: the stability of the melt pool, which was captured using our proposed temporal features, and the stability of printer control, which we aimed to improve by reducing the number of times the laser settings needed to be changed in real-time. This latter stability criterion was addressed through our contribution of pore density prediction. For both contributions, we saw increased accuracy in detecting porosity, with the combination of the two showing the strongest correlations with the measured pore densities. For the temporal features, improvements were most noticeable in the laser speed and keyhole pore density predictions. This may be a result of capturing additional dynamic information using the temporal features. Clearly, the time dimension is already present in the definition of laser speed, and as that speed increases, the laser will cover more area and could interact with a greater variety of powder conditions (e.g., different powder sizes, impurities, and temperatures). We suspect that this greater variety could be increasing the temporal variance in the video features. Similarly, keyhole pore creation is known to be associated with a variety of melt pool fluctuations (e.g., vaporisation of volatile elements and keyhole collapse) [44]. Our temporal features are likely capturing some of that dynamic behaviour. For pore density prediction, we observed improved results mainly because some off-nominal laser settings did not introduce porosity. By predicting pore densities, our LPBF monitoring system was able to ignore these inconsequential laser parameter changes.

Our two main contributions, the temporal features and pore density predictions, are novel and, to our knowledge, have not been applied to LPBF monitoring before. With respect to the use of temporal features, simple temporal denoising methods have been applied [8,15], but apart from that, the time dimension has largely been ignored in LPBF monitoring. Our results suggest that this temporal denoising might actually be counterproductive: it may filter out the dynamic melt pool behaviour that we found useful for detecting porosity. With respect to pore density prediction, the closest related works focus on the prediction of laser parameters such as laser speed and power [10,13,26]. The hypothesis behind such an approach is that the melt pool should always *look* like it was created with the same nominal laser parameters. Our results here show that such an approach is over-conservative: it would recommend that laser parameters be changed even when no porosity is being introduced. Overall, our combination of temporal features and pore density estimation provides the most accurate prediction of porosity, resulting in more accurate signals to control part quality and to reduce scrap rates.

The monitoring performed in this work was based on high-speed imaging. The sampling rate of 20,000 fps was chosen as a trade-off between what is needed to observe pore creation and what video speed can possibly be processed in real-time. The creation of pores in an LPBF process occurs over very short time periods, typically less than 100 ms [44]. Thus, a high sampling rate is required to even observe the pore creation phenomena and the events leading up to it. At the same time, the multiple video analysis steps presented here require computation time, a resource that will be limited when extending this work to a real-time system. While we are currently working on a real-time version of this monitoring system, the camera’s high sampling rate has already led us to make design decisions that reduce the amount of computations required. These design decisions include the avoidance of deep learning, placing limits on the number of video features (e.g., the spatter direction features were limited to six even through Ji and Han used 80 spatter direction features in a similar approach [38]) and the preference for smaller neural networks for pore density prediction.

On a related note, we acknowledge that the results presented here consider a single set of spatial features, one which is not comprehensive. In this work, we chose spatial features with two criteria in mind. First, we wanted to test features that capture a variety of information, including information on the melt pool, spatters, and the image as a whole (e.g., image texture). Second, we selected features that we believe can be computed in real-time. It may be the case that the improvements seen with the proposed contributions would differ in magnitude depending on the spatial features used. Deep learning features in particular may be able to increase pore prediction accuracy simply through using more descriptive image features. Nevertheless, we hypothesise that the proposed stability-related contributions would show improvements in pore detection regardless of the features used. It remains part of our future work to test this hypothesis.

Despite these improvements in pore density prediction, we did recognise a couple of limitations. First, the temporal features seemed to influence laser speed and keyhole pore density predictions more so than laser power and lack-of-fusion pore density predictions. For lack-of-fusion pore density prediction specifically, this limited improvement may be due to the temporal features’ sensitivity to laser stopping and starting effects at the beginning and end of each scan line. This sensitivity is notable in Figure 7c, where high lack-of-fusion pore density predictions are seen around the border of the object even when no porosity is seen. A simple workaround may be to ignore the temporal features when performing predictions near the object border, thus eliminating the laser stopping and starting effects at the expense of poorer pore predictions around the object borders. Meanwhile, we may be seeing limited improvement in laser power estimation simply because existing spatial features are already able to predict it quite well [12,13,24].

A second limitation of this work is related to the measurement of pore densities from the CT image. We assume that all pores on or below an off-nominal print layer have the same label. In practice, this assumption can be challenging to apply. To improve the suitability of this assumption, we deliberately ignored the object border regions. These border regions show porosity for a variety of reasons that are beyond our control (e.g., geometry, head dissipation, laser stopping and starting effects) [45], making it challenging to determine what types of pores are present there. Similarly, we made sure to introduce a gap of nominally printed layers between the off-nominal layers we used to create pores. While our off-nominal laser settings were chosen carefully to introduce a specific pore type, there is the possibility that the occasional pore of a different type could have been created in nearby nominally printed layers. Based on our reported improvements in pore prediction accuracy, we suspect that any incorrectly labelled pores were small enough in number not to significantly impact our results. Nevertheless, it should be noted that the labelling of the pore type can be both challenging and can influence the results of the proposed LPBF monitoring platform. This labelling should be carefully considered when training the neural network.

Finally, we acknowledge that the experimentation presented in this study is not comprehensive. Only a single test object was used for experimentation, and it had a rather simple geometry. It has been established that part geometry can have an impact on the resulting porosity [46], with more complex geometries inducing greater porosity. It remains to be seen how the proposed LPBF monitoring contributions perform across different printed objects with more complex geometries. This too is planned future work. Nevertheless, we showed a proof-of-concept experiment in which pore prediction was improved through the explicit encoding of LPBF process stability into the monitoring system.

## 5. Conclusions

We proposed herein that LPBF monitoring systems could predict porosity better by explicitly measuring LPBF stability. In particular, we proposed two ways of capturing that stability using high-speed camera data. First, temporal video features were introduced to compute the variance in the LPBF process across time. Second, we proposed a machine learning model for the direct prediction of pore densities. Our experiments showed that these contributions improved the estimation of real pore creation by both capturing dynamic LPBF behaviour and directly learning its relationship with the resulting porosity. The temporal features improved the prediction of keyhole pores by over 20%, while the direct prediction of pore densities improved prediction accuracy by 24–42%. These results are an encouraging first step for the encoding of stability measures in LPBF monitoring. Future work will look to confirm these results over different part geometries and different image feature sets.

## Figures and Tables

**Figure 1 sensors-22-03740-f001:**
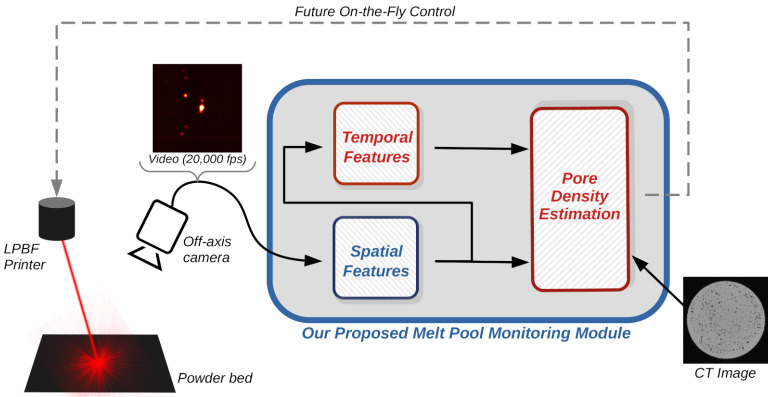
Flowchart of the proposed LPBF monitoring system with our main contributions highlighted in red. A high-speed video camera is used to capture melt pool and spatter information, which is then summarised into spatial *and temporal* features. These features are then used to *directly predict* the density of pores created in the 3D-printed part. See the text for further details.

**Figure 2 sensors-22-03740-f002:**
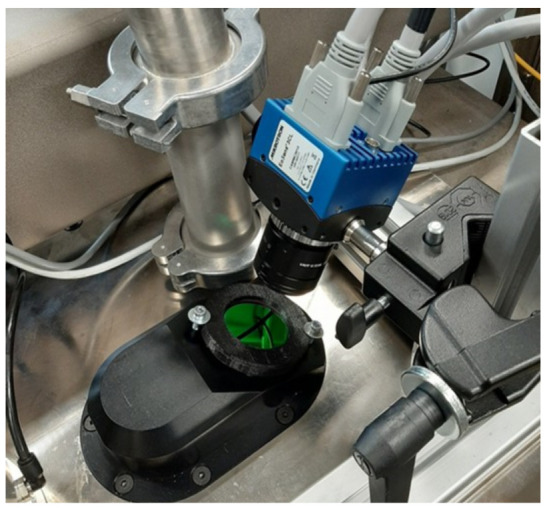
Installation of the high-speed camera on our LPBF printer.

**Figure 3 sensors-22-03740-f003:**
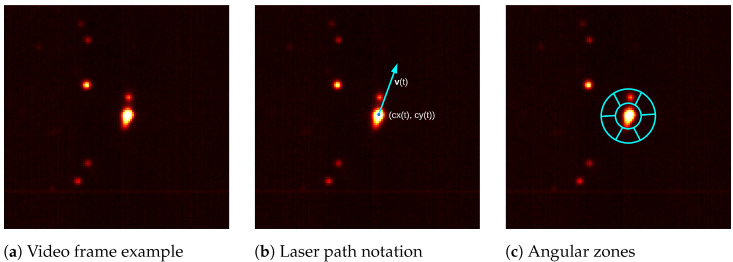
An example frame of the high-speed video along with notations used in the feature extraction. The frame alone 𝔗(·,·,t) is shown in (**a**), while the laser location (cx(t),cy(t)) and direction v(t) are shown in blue in (**b**). The angular zones used for measuring spatter direction are shown in blue in (**c**).

**Figure 4 sensors-22-03740-f004:**
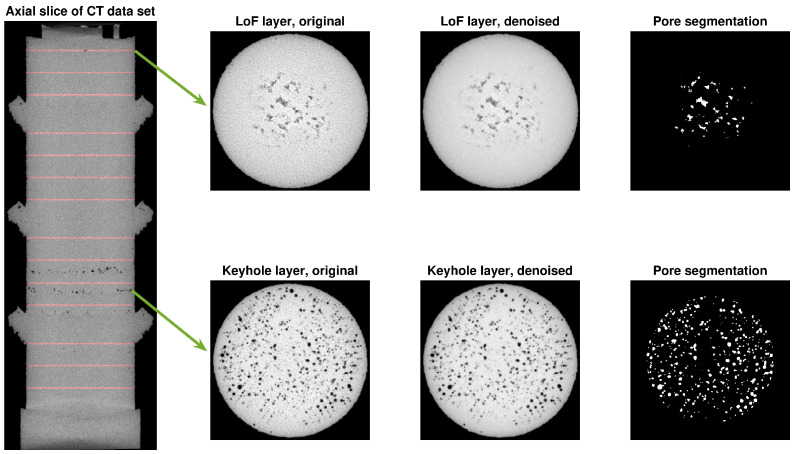
Illustration of the pore segmentation process. Shown is a trans-axial (vertical) slice through the centre of a printed object, with the locations of the non-optimal print layers indicated in red (**left**). Horizontal slices through a lack-of-fusion layer (**top**) and a keyhole layer (**bottom**), indicated with arrows, are shown. For these layers, the original CT, the denoised CT, and the segmented pores are shown from left to right.

**Figure 5 sensors-22-03740-f005:**
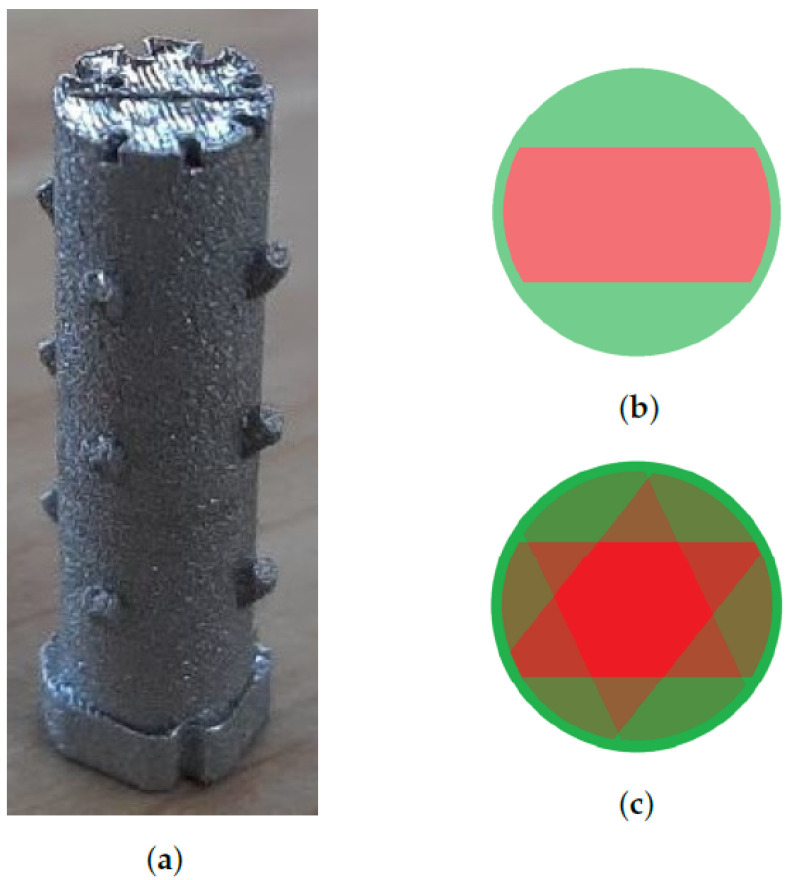
The object printed to generate training and testing datasets (left). Pores were induced in the bulk of the object over groups of three consecutive layers using the printing schemes on the right. (**a**) The object after printing and cutting. (**b**) Off-nominal print layer (red = area with off-nominal laser settings). (**c**) Three consecutive off-nominal print layers.

**Figure 6 sensors-22-03740-f006:**
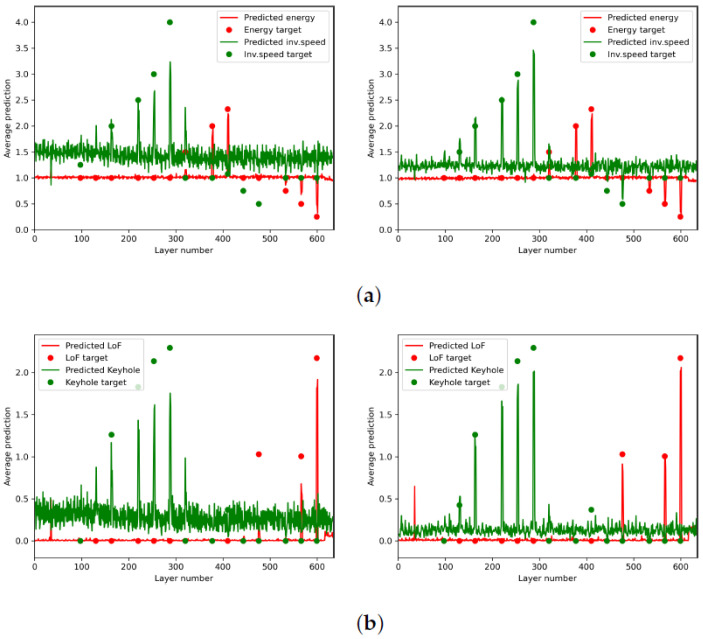
The predictions on the test data set averaged per layer, with the targets at the off-nominal layers indicated with dots. (**a**) System settings’ predictions without (left) and with (right) temporal features. (**b**) Pore density predictions without (left) and with (right) temporal features.

**Figure 7 sensors-22-03740-f007:**
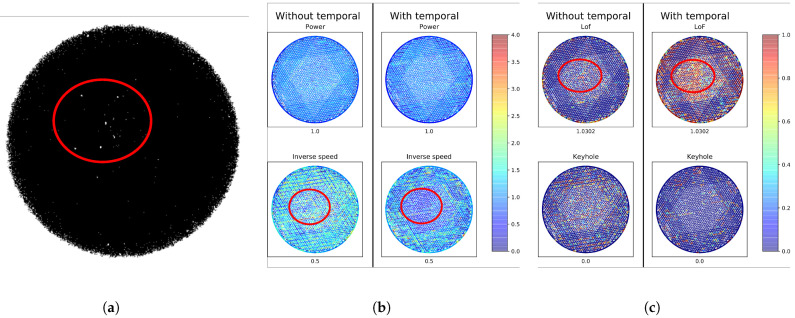
CT image, predicted laser settings and predicted pore densities, for a layer printed with higher-than-nominal laser speed (Section ID = 11 in Table 1). Note that the addition of temporal features improved the predictions for both laser speed and lack-of-fusion porosity. These improved predictions were most noticeable in the region highlighted in red. (**a**) CT image with lack-of-fusion pores. (**b**) Predicted laser settings. (**c**) Predicted pore densities.

**Figure 8 sensors-22-03740-f008:**
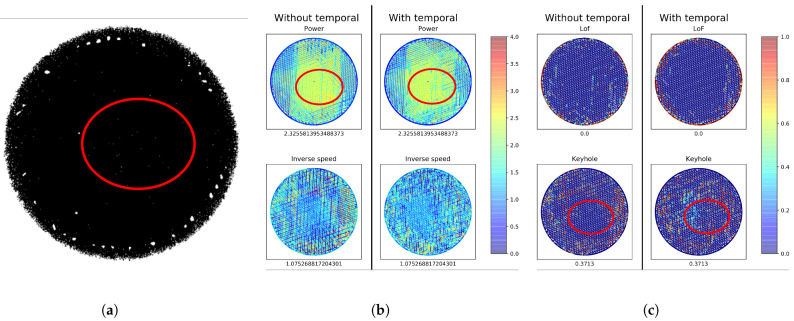
CT image, predicted laser settings, and predicted pore densities, for a layer printed with higher-than-nominal laser power (Section ID = 11 in Table 1). Note that the laser settings’ predictions would recommend a larger amount of intervention in the printing process, especially in the region highlighted in red, despite the limited production of pores. (**a**) CT Image with keyhole pores. (**b**) Predicted laser settings. (**c**) Predicted pore densities.

**Table 1 sensors-22-03740-t001:** The laser power and speed as a percentage of the nominal values and the lack-of-fusion and keyhole densities (computed from the CT scan) for each off-nominal section.

Section	Laser Power	Laser Speed	Lack-of-Fusion	Keyhole
ID	(% of Nominal)	(% of Nominal)	LPD	LPD
1	100	80.0	0	0
2	100	66.7	0	0.691
3	100	50.0	0	2.94
4	100	40.0	0	4.82
5	100	33.3	0	5.93
6	100	25.0	0	6.52
7	150	100	0	0
8	200	100	0	0
9	233	93.0	0	0.575
10	100	133	0	0
11	100	200	1.77	0
12	75.0	100	0	0
13	50.0	100	1.71	0
14	25.0	100	4.77	0

**Table 2 sensors-22-03740-t002:** The high-speed video features used as inputs to the pore density estimation.

Feature Type	Number of Features	Described in...
Melt pool area	1	Section 2.2.1
Melt pool width–length ratio	1	Section 2.2.2
Amount of spatter	1	Section 2.2.3
Number of spatters	1	Section 2.2.4
Melt pool intensity	1	Section 2.2.5
Spatter direction	6 (1 per angular zone)	Section 2.2.6
Histogram of oriented gradients	9 (1 per histogram bin)	Section 2.2.7
Temporal variances	20 (1 per spatial feature)	Section 2.3

**Table 3 sensors-22-03740-t003:** Correlation coefficients between predicted and ground truth outputs on the test set, for scenarios with and without temporal features.

	Ground Truth Correlations		
Predicted Measure	Without Temporal	With Temporal	Improvement
	Features	Features	
Laser Power	0.871	0.892	+0.021
Inverse Laser Speed	0.645	0.813	+0.168
Lack-of-Fusion LPD	0.881	0.916	+0.035
Keyhole LPD	0.650	0.819	+0.169

**Table 4 sensors-22-03740-t004:** Coefficient of (multiple) correlations between predicted variables (using temporal features) and pore densities on the test set.

Predictor	Correlation with Porosity	
	Lack-of-Fusion	Keyhole
Laser Speed and Power	0.524	0.616
Lack-of-Fusion LPD	0.916	-
Keyhole LPD	-	0.819
Improvement	+0.392	+0.203

## Data Availability

The data presented in this study are available upon reasonable request from the corresponding author.

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
