# Peer review of "Encoding Stability into Laser Powder Bed Fusion Monitoring Using Temporal Features and Pore Density Modelling"

_sensors, 2022, doi:10.3390/s22103740_

Round 1

Reviewer 1 Report

This manuscript presents a video based LPBF process monitoring method that incorporates both spatial and temporal features to quantify printing stability. A neural network regressor is developed to predict pore densities. Experimental results of SS316L specimens validate that after incorporating temporal features, the prediction accuracy of keyhole pore densities and laser speed is significantly improved. The research topic is highly relevant to LPFB process monitoring and this manuscript is well organized. The following questions should be addressed before publication.

A very high image sampling rate of 20000 fps is adopted, which may cause difficulties in real time monitoring. Please discuss on this issue. In addition, is such high sampling rate necessary for offline/real time monitoring?

20 spatial features are selected in this work. All of them should be listed. On what basis are these 20 spatial features as well as the corresponding temporal features selected for neural network inputs?

Table 2 shows that temporal features significantly improve the prediction for laser speed and keyhole pore density, but marginally improvements are observed for laser power and LoF pore density. What are the underlying reasons for the different influences of temporal features?

The quality of Figure 2 should be improved.

Typo in lines 416-417: “as laser speed contains contains…”

Reviewer 2 Report

This paper focused on explicitly encoding stability into LPBF monitoring systems through the use of temporal features and pore density modelling.  A high-speed video camera was used to collect data. For experimental validation, 316 stainless steel part was also printed. The stability was considered in two respects in the proposed methodology: the stability of the melt pool which was captured using the proposed temporal features, and the stability of printer control which was improved by reducing the number of times laser settings needed to be changed in real time. The reviewer understands the novelty and experimental findings of this paper. However, there are some questions/comments. The reviewer asks the authors to reply and modify the paper following the comments.

-There are typos and grammatical errors all over the manuscript which need to be addressed; It is highly recommended that the authors enhance the language of the entire manuscript.

-In section 22, figure 3c, how did the authors choose the angular grid for measuring spatter direction? it should be clarified and explained while comparing with other works.

-In section 2.6.2, line 314-315, the authors mentioned “To avoid border effects, pore densities were computed only on the central cylindrical core of the object (radius = 2 mm)”.

Can you explain how borders can affect the total pore densities? If we consider the border effect, what parameters will change?

-In section 3.2, line 390-391, the authors mentioned: “We hypothesized that predicting pore densities, we would obtain signals that more strongly correlate with part quality than predicting a printer’s laser settings.” The sentence is not clear and needs to be re-written.

-Line 400, one ‘porosity’ is redundant in “The CT image in Fig. 8a shows no porosity porosity within the red oval in this layer”.

-Line 416-417, one ‘contain’ is extra in “which makes sense as laser speed contains contains the time dimension in its measure”

-Line 419-420 the sentence “For pore density prediction, improvements we mostly seen by ignoring laser parameter changes that did not introduce porosity.” is vague; needs to be re-written.

- In discussion part, the authors should compare their method and outcomes with other relevant methods. They should precisely highlight the benefit and novelty of using their method compared to other works.
